METHODS AND RESOURCES

# Systematic and scalable genome-wide essentiality mapping to identify nonessential genes in phages

**Denish Piya[1], Nicholas Nolan[2], Madeline L. Moore[3], Luis A. Ramirez Hernandez[3], Brady F. Cress[1,4], Ry Young[5], Adam P. Arkin** ◉[1,2,3]*, **Vivek K. Mutalik** ◉[1,3]*

**1** Innovative Genomics Institute, University of California-Berkeley, Berkeley, California, United States of America, **2** Department of Bioengineering, University of California-Berkeley, Berkeley, California, United States of America, **3** Environmental Genomics and Systems Biology Division, Lawrence Berkeley National Laboratory, Berkeley, California, United States of America, **4** Department of Molecular and Cell Biology, University of California-Berkeley, Berkeley, California, United States of America, **5** Department of Biochemistry and Biophysics, Center for Phage Technology, Texas A&M University, College Station, Texas, United States of America

* aparkin@lbl.gov (APA); vkmutalik@lbl.gov (VKM)

**Data Availability Statement:** All relevant data are within the paper and its Supporting Information files. The underlying data for all figures are provided in supporting information file S1 Data.

## Abstract

Phages are one of the key ecological drivers of microbial community dynamics, function, and evolution. Despite their importance in bacterial ecology and evolutionary processes, phage genes are poorly characterized, hampering their usage in a variety of biotechnological applications. Methods to characterize such genes, even those critical to the phage life cycle, are labor intensive and are generally phage specific. Here, we develop a systematic gene essentiality mapping method scalable to new phage–host combinations that facilitate the identification of nonessential genes. As a proof of concept, we use an arrayed genome-wide CRISPR interference (CRISPRi) assay to map gene essentiality landscape in the canonical coliphages λ and P1. Results from a single panel of CRISPRi probes largely recapitulate the essential gene roster determined from decades of genetic analysis for lambda and provide new insights into essential and nonessential loci in P1. We present evidence of how CRISPRi polarity can lead to false positive gene essentiality assignments and recommend caution towards interpreting CRISPRi data on gene essentiality when applied to less studied phages. Finally, we show that we can engineer phages by inserting DNA barcodes into newly identified inessential regions, which will empower processes of identification, quantification, and tracking of phages in diverse applications.

## Introduction

Bacteriophages (phages) are the most abundant biological entities on earth and are postulated to play a crucial role in environmental nutrient cycles, agricultural productivity, and human health [1,2]. The full scope of the roles phages play in regulating the activity and adaptation of microbial communities is still emerging [3–5]. Phages represent one of the largest pools of genetic diversity with unexplored functional information [6–9]. For example, the majority of

The complete data from phage Barseq experiments are deposited here https://doi.org/10.6084/m9.figshare.22817084.

**Funding:** This material by ENIGMA- Ecosystems and Networks Integrated with Genes and Molecular Assemblies (http://enigma.lbl.gov), a Science Focus Area Program at Lawrence Berkeley National Laboratory is based upon work supported by the U.S. Department of Energy, Office of Science, Office of Biological & Environmental Research under contract number DE-AC02-05CH11231 (to V.K.M. and A.P.A.). Phage P1 assay part of this research was supported by the U.S. Department of Energy, Office of Science, through the National Virtual Biotechnology Laboratory, a consortium of U.S. Department of Energy national laboratories focused on response to COVID-19, with funding provided by the Coronavirus CARES Act (to V.K.M. and A.P.A.) and the early design part of this project was funded by the Microbiology Program of the Innovative Genomics Institute, Berkeley (to A.P.A. and V.K.M.). R.F.Y acknowledges funding from the National Institute of General Medical Sciences (NIGMS) grant R35GM136396. The funders had no role in study design, data collection and analysis, decision to publish, or preparation of the manuscript.

**Competing interests:** V.K.M., D.P. and A.P.A. are holders of a patent (pending) on the phage barcoding technology. V.K.M. is a co-founder of Felix Biotechnology. A.P.A. is a co-founder of Boost Biomes and Felix Biotechnology. A.P.A. is a shareholder in and advisor to Nutcracker Therapeutics. The remaining authors declare no competing interests.

**Abbreviations:** aTc, anhydrotetracycline; CRISPRi, CRISPR interference; crRNA, CRISPR RNA; EOP, efficiency of plating; Lpa, Late Promoter Activator; ORF, open reading frame; PAM, protospacer adjacent motif.

phage genes (>70% to 80%) identified by bioinformatic analysis are of unknown function and show no sequence similarity to characterized genes [10]. Homology-based approaches to connect phage genes to their function are limited by the lack of experimental data [11,12]. While focused biochemical and genetic analysis are the gold standard for assessment of gene functions, most of these methods are not scalable to the vast amount of new genes being discovered [10]. Unless we develop methods to fill the knowledge gap between phage genetic diversity and gene function, we will be seriously constrained in understanding the mechanistic ecology of phages in diverse microbiomes and harness them as engineerable antimicrobials and microbial community editors [13,14].

Gaps in phage gene-function knowledge exist even for some of the most well-studied canonical phages [15,16]. Nevertheless, the application of classical phage genetic tools to a few canonical phages over the last few decades has paved the way for generating foundational knowledge of the phage life cycle [15,17,18]. A number of recent technological innovations have also addressed the growing knowledge gap between phage-gene-sequence and the encoded function [19–21]. These innovations range from classical recombineering methods [22,23] and new phage engineering platforms [24–28] to genome editing tools such as CRISPR systems, with or without recombineering technology to create individual phage mutants [13,25,29–33]. Importantly, no method for assessing essentiality without genome modification has been reported. As such, the field is in need of genome-wide technologies that can be used rapidly across diverse phages to assess gene function [14]. At minimum, such a method would provide the foundational knowledge of which phage genes are essential for its infection cycle in a given host, a prerequisite for understanding host range and for engineering.

Catalytically inactive CRISPR RNA (crRNA)-directed CRISPR endonucleases or CRISPR interference (CRISPRi) technology has emerged as a facile tool for carrying out genome-scale targeted interrogation of gene function in prokaryotic and eukaryotic cells without modification of the genome [34,35]. A catalytically inactive or "dead" Cas protein (such as dCas9 or dCas12a) enables programmable transcriptional knockdown (by binding to DNA and forming a transcriptional road block) yielding a loss-of-function phenotype in a DNA sequence–dependent manner [36–41]. While CRISPRi was first developed using dCas9, alternative Cas variants like dCas12a have achieved efficient knockdown in diverse bacteria [42–44]. Both dCas9 and dCas12a are similarly effective for CRISPRi in many circumstances; however, advantages of dCas12 include more efficient restriction of a covalently modified phage genome (for example, T4 phage [45]) and simpler cloning of dual crRNAs on short oligos relative to longer dCas9 single-guide RNAs. Recent work demonstrated that dCas12a is capable of inhibiting infection by phage λ when targeting the essential gene *cro*, suggesting that application of dCas12a with arrayed crRNAs might facilitate genome-wide fitness measurements in phages [46]. The ability to effectively block transcription at target sites distant from promoters makes dCas12a potentially well suited for repressing transcription of phage genes within operons that show overlapping genetic architecture [15,17,18,47,48] and those that are highly regulated or vary in expression levels [49–51] in a noncompetitive plaque assay.

Here, we adopted catalytically inactive Cas12a (dCas12a) to carry out systematic genome-wide interference assays in 2 canonical phages. The first is coliphage lambda, arguably the best characterized virus in terms of individual gene function and developmental pathways [17]. The second is coliphage P1, which, as a powerful generalized transducing phage, was instrumental in the development of *Escherichia coli* as a primary genetic model [52]. Its genome is also well annotated but less experimentally characterized than lambda. We first benchmark the CRISPRi technology by applying it to a known set of essential and nonessential genes in both phages (lytic variants, λcI857, and P1vir, here onwards as lambda or λ and P1, respectively) and then extend it genome-wide to query essentiality of all genes in both phages. Although

some ambiguities are revealed and significant polarity effects are detected, the method is clearly demonstrated to be applicable to the rapid assignment of nonessential loci in phages, thus paving the way for systematic genome-scale engineering in a variety of applications.

## Results

### Setting up CRISPRi assay targeting phage genes

To ascertain that dCas12a can repress phage gene expression, we designed a phage targeting CRISPRi plasmid system following earlier work [53] by expressing both dCas12a and a crRNA to target specific genes (Methods). Briefly, we placed dCas12a under an anhydrotetracycline (aTc)-inducible Tet promoter and the CRISPR array including the phage targeting crRNA under a strong constitutive promoter on a medium copy plasmid. We then selected a set of known essential and nonessential genes that encode proteins needed at different copy numbers for lambda and P1 (Fig 1). For lambda, we chose *E*, which encodes the major capsid protein, and *Nu1*, which encodes the small terminase subunit. For P1, we chose genes *23*, *pacA*, and *sit*, encoding the major capsid protein, large terminase subunit, and tape measure protein, respectively [17,52]. In addition to these essential phage genes, we also chose nonessential P1 genes such as *ppp*, *upfB*, or *ddrB* [54]. We identified Cas12a protospacer adjacent motif (PAM) sites (TTTV) in the 5′ end of the genes (approximately 20% downstream of the start site) and used 28 bp nucleotide sequence immediately downstream of the PAM site in the coding strand as the spacer region for designing crRNAs.

We performed plate-based CRISPRi efficiency assays by moving each variant of the CRISPRi plasmid into *E. coli* BW25113 separately and induced the expression of dCas12a before plating serial dilution of the 2 phages (Methods). After overnight incubation, we compared the plating efficiency on lawns expressing the gene-targeting crRNAs versus a control lawn in which the crRNA did not target either phage (Fig 1B). We observed that induction of CRISPRi targeting essential genes *E* and *nu1* of lambda and *mcp*, *pacA*, and *sit* of P1 all showed severe compromise in phage growth (measured as plaque formation), whereas targeting nonessential genes *ppp*, *upfB*, or *ddrB* of P1 did not. Overall, our CRISPRi benchmarking assays indicated that the dCas12a CRISPRi platform can be used to assess essentiality of phage genes expressed at different levels during the infection cycle.

### Genome-wide CRISPRi to map gene essentiality in λ

To extend our initial observations to systematically query gene essentiality at genome-wide levels, we considered λ as our pilot case, since it is the most deeply characterized phage with detailed assessments of gene functions well represented in the literature [17,55]. Decades of work on suppressible nonsense mutants of λ phage have helped to define 28 genes (out of total 73 open reading frames (ORFs)) as essential for phage growth (Table 1) providing a well-characterized test bed for validation of our genome-wide CRISPRi assay.

We designed individual crRNAs targeting 67 out of 73 genes of the lambda genome, using the same criteria as used for the pilot studies (by locating PAM sites in the 20% to 33% of the way through the CDS region of each gene to account for any possible alternative start sites for genes) (S1 Fig). The remaining 6 genes (*cII*, *ninD*, *ninE*, *ninH*, *Rz1*, and lambdap35) were not tested here due to lack of canonical PAM sites. The designed crRNAs were synthesized as separate pairs of oligos and cloned into the CRISPRi plasmid system downstream of a strong constitutive promoter (Methods). Each of these plasmids encoding crRNAs were arranged in an arrayed format and moved into *E. coli* BW25113 as indicator strains for the plate-based CRISPRi assay to measure the efficiency of plating (EOP) (Fig 2, described above, and Methods). The EOP is a quantitative measure of the knockdown for each guide RNA. We assessed the

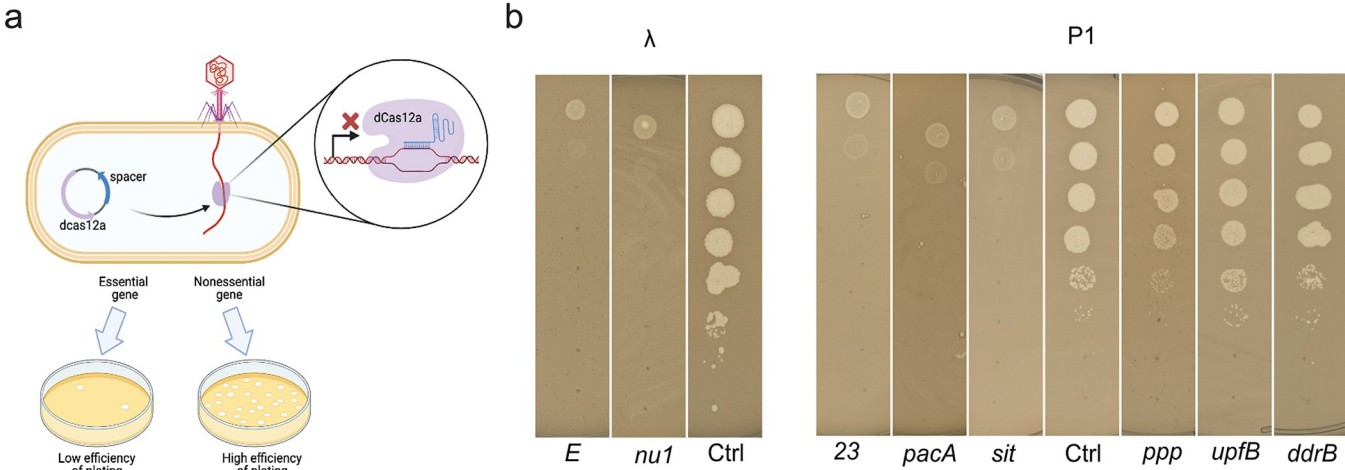

**Fig 1. Design and testing of CRISPRi knockdowns to assess gene essentiality in phages lambda and P1.** (**a**) Schematic of CRISPRi assay system. (**b**) Representative images of plaque assays to validate the dCas12a CRISPRi system using gene targets with known essentiality. We employed crRNAs targeting 2 essential genes of phage λ: genes encoding major capsid protein (E) or DNA packaging subunit (Nu1). For phage P1, we used crRNA targeting 3 essential genes: encoding the major capsid protein (gene *23* encoding Mcp), DNA packaging subunit (PacA), and tape measure protein (Sit); and 3 nonessential genes: *ppp*, *upfB*, or *ddrB*. For comparison, phage plaques appearing on an *E. coli* BW25113 lawn expressing a nontargeting crRNA as a control are shown for both phages (Ctrl).

reproducibility of EOP estimations by carrying out biological replicates (total assays >150) and depicted the average EOP of every gene on the lambda map (Fig 3 and Table 1, Note A in S1 Text).

In total, our CRISPRi assays indicated 35 genes as essential and 32 genes as nonessential. For example, consistent with the literature [17], knockdown of genes that encode factors involved in the structural assembly of λ virions, either the capsid morphogenesis (Nu1, A, B, C, Nu3, D, E, FI) or tail morphogenesis (V, G, G-T, H, M, L, K, I, J), were detrimental to phage growth with 5-log reduction in EOP. Similarly, repression of genes encoding crucial factors involved in the lytic phase of lambda phage growth cycle, such as transcription antiterminators (proteins N and Q), DNA replication (proteins O and P), transcriptional regulator (Cro), and programmed disruption of host membrane (holin/antiholin S and S') all showed approximately 4- to 5-log reduction in EOP, indicating their important role in phage fitness phenotype (Table 1).

The longest stretch of dispensable DNA for lambda encompasses >30% of its genome and is made up of 4 clusters of genes arranged between gene *J* and gene *N* (Fig 3). These include a cluster of genes *lom-stf-tfa*, 20 genes within pL operon, genes in the immunity region (*rex* and *cI* genes), and genes encoding the lysis program (*R* and *Rz*). We found, except for gene *N*, all genes within pL operon are dispensable for lambda plaque formation (Fig 3 and Table 1). Some of these genes provide functions that would not be expected to have a plaque-formation defect on fully competent lawns, like the superinfection exclusion genes (*rexA*, *rexB*, *sieB*) [56] and genes involved in lysogeny (*int*, *xis*, *CIII*) [57], but others might, such as homologous recombination (*exo*, *bet*, *gam*) [58] and inhibition of host cell division (*kil*) [59]. The knockdown of *ral* (encoding a restriction inhibitor protein) does not result in a major defect in the EOP because our indicator strain lacks a functional type I restriction system [60,61]. To probe the essentiality of *ral*, we repeated the knockdown assays on 2 different strains with active type I restriction system (Methods). These assays indicate the conditional essentiality of *ral* that depends on the genotype of the target bacterial strain (S2 Fig). The dispensability of the side tail fiber (which requires *stf* and *tfa*) is in agreement with the known frameshift mutation in the *stf* locus in laboratory strains of λ [62]

**Table 1. Gene essentiality mapping of phage lambda genome.**

| Locus_tag | Gene | function/Protein name [17] | EOP_average | S.D. | This work | Literature |
|---|---|---|---|---|---|---|
| lambdap01 | nu1 | DNA packaging protein | 5.5E-06 | 2.1E-06 | E | E [91] |
| lambdap02 | A | TerL | 1.3E-06 | 4.7E-07 | E | E [92,93] |
| lambdap03 | W | gpW family protein | 3.0E-05 | 1.4E-05 | E | E [93,94] |
| lambdap04 | B | portal | 2.5E-05 | 2.1E-05 | E | E [92,93] |
| lambdap05 | C | S49 family peptidase/capsid component; Viral protease | <2.0E-7 | | E | E [93,95] |
| lambdap06 | nu3 | scaffolding protein | <2.0E-7 | | E | E [93,96] |
| lambdap07 | D | head decoration protein | 6.8E-06 | 2.4E-07 | E | E [93,97] |
| lambdap08 | E | major capsid protein | <6.4E-6 | | E | E [93,97] |
| lambdap09 | Fi | DNA packaging protein FI | 2.0E-05 | 0.0E+00 | E | E [93,98] |
| lambdap10 | Fii | head-tail joining protein | <2.0E-7 | | E | E [94] |
| lambdap11 | Z | tail protein | 2.5E-05 | 7.1E-06 | E | E [55] |
| lambdap12 | U | tail protein | 3.5E-05 | 2.1E-05 | E | E [55] |
| lambdap13 | V | tail protein | 1.7E-05 | 1.9E-05 | E | E [55] |
| lambdap14 | G | minor tail protein G | <6.4E-6 | | E | E [99] |
| lambdap15 | T | tail assembly protein T | 1.2E-05 | 1.2E-05 | E | E [99] |
| lambdap16 | H | tail tape measure protein | 1.0E-06 | 9.4E-07 | E | E [99] |
| lambdap17 | M | tail protein | <2.0E-7 | | E | E [99] |
| lambdap18 | L | minor tail protein L | <2.0E-7 | | E | E [99] |
| lambdap19 | K | tail protein | 8.7E-05 | 1.2E-04 | E | E [99] |
| lambdap20 | I | tail component | 2.6E-05 | 2.7E-05 | E | E [100] |
| lambdap21 | J | host specificity protein J | 1.1E-05 | 1.2E-05 | E | E [92] |
| lambdap26 | lom | Outer membrane beta-barrel protein Lom | 8.8E-01 | 5.3E-01 | NE | NE [17] |
| lambdap27 | stf | protail fiber N-terminal domain containing protein | 3.0E+00 | 2.8E+00 | NE | NE [62] |
| lambdap90 | orf206b | hypothetical protein | 4.3E+00 | 1.1E+00 | NE | NE [62] |
| lambdap28 | tfa | tail fiber protein | 6.3E+00 | 5.3E+00 | NE | NE [62] |
| lambdap29 | orf-194 | tail fiber assembly protein | 3.3E+00 | 1.1E+00 | NE | NE [62] |
| lambdap80 | ea47 | | 5.5E+00 | 6.4E+00 | NE | NE [17] |
| lambdap81 | ea31 | | 1.2E+00 | 1.1E+00 | NE | NE [17] |
| lambdap82 | ea59 | ATP-dependent endonuclease | 1.3E+01 | 1.7E+01 | NE | NE [17] |
| lambdap33 | int | tyrosine-type recombinase/integrase | 2.8E+00 | 3.2E+00 | NE | NE [55] |
| lambdap34 | xis | excisionase | 1.6E+00 | 1.1E+00 | NE | NE [55] |
| lambdap35 | | | | | NT | |
| lambdap36 | ea8.5 | | 1.4E+00 | 1.3E+00 | NE | NE [17] |
| lambdap83 | ea22 | ead/Ea22-like family protein | 2.2E+00 | 2.3E+00 | NE | NE [17] |
| lambdap37 | orf61 | hypothetical protein | 9.7E-01 | 8.1E-01 | NE | NE [17] |
| lambdap38 | orf63 | DUF1382 family protein | 1.3E+00 | 3.8E-01 | NE | NE [17] |
| lambdap39 | orf60a | DUF1317 domin-containing protein | 2.4E+00 | 2.0E+00 | NE | NE [17] |
| lambdap41 | exo | YqaJ viral recombinase family protein | 6.9E-01 | 4.4E-01 | NE | NE [17] |
| lambdap84 | bet | recombination protein Bet | 2.1E+00 | 2.5E+00 | NE | NE [17,101] |
| lambdap42 | gam | host-nuclease inhibitor protein Gam | 1.4E+00 | 1.3E+00 | NE | NE [17,101] |
| lambdap85 | kil | host cell division inhibitory peptide Kil | 1.0E+00 | 0.0E+00 | NE | NE [102] |
| lambdap86 | cIII | protease FtsH-inhibitory lysogeny factor CIII | 1.1E+00 | 3.1E-01 | NE | NE [103,104] |
| lambdap45 | ea10 | DUF2528 family protein | 1.7E+00 | 4.7E-01 | NE | NE [17] |
| lambdap46 | ral | Restriction inhibitor protein ral | 1.1E+00 | 3.2E-01 | NE | NE [60,61] |
| lambdap47 | orf28 | hypothetical protein | 1.3E+00 | 4.7E-01 | NE | NE [17] |
| lambdap48 | sieB | Superinfection exclusion protein B | 1.5E+00 | 7.1E-01 | NE | NE [105] |
| lambdap49 | N | antitermination protein N | 1.8E-04 | 1.2E-04 | E | E [106,107] |

*(Continued)*

**Table 1.** (Continued)

| Locus_tag | Gene | function/Protein name [17] | EOP_average | S.D. | This work | Literature |
|---|---|---|---|---|---|---|
| lambdap53 | *rexB* | exclusion protein | 1.4E+00 | 1.3E+00 | NE | NE [17,105] |
| lambdap87 | *rexA* | exclusion protein | 3.8E+00 | 4.0E+00 | NE | NE [17,105] |
| lambdap88 | *cI* | lysogenic repressor | 5.3E+00 | 6.6E+00 | NE | NE [103,108,109] |
| lambdap57 | *cro* | lytic repressor | <7.5E-6 | | E | E [110] |
| | *cII* | | | | NT | NE [103,108,109] |
| lambdap89 | *O* | replication protein | <7.5E-6 | | E | E [111,112] |
| lambdap61 | *P* | DNA replication protein | <7.5E-6 | | E | E [111,112] |
| lambdap62 | *ren* | protein ren | 1.20E-02 | 7.7E-06 | E | NE [113] |
| lambdap63 | *ninB* | recombination protein NinB | 3.00E-05 | 5.4E-04 | E | NE [17,65] |
| lambdap64 | *ninC* | phosphoadenosine phosphosulfate reductase family protein | clearing on E-1 | | E | NE [17,65] |
| | *ninD* | | | | NT | NE [17,65] |
| | *ninE* | | | | NT | NE [17,65] |
| lambdap67 | *ninF* | | 4.00E-05 | 4.0E-06 | E | NE [17,65] |
| lambdap68 | *ninG* | recombination protein NinG | 8.0E-06 | 1.1E-04 | E | NE |
| | *ninH* | | | | NT | NE [17,65,113] |
| lambdap70 | *ninI* | serine/threonine phosphatase | 4.0E-06 | 5.8E-06 | E | NE [17,65] |
| lambdap71 | *Q* | antitermination protein | 1.6E-05 | 6.1E-06 | E | E [65] |
| lambdap73 | *orf-64* | hypothetical protein | 1.2E-02 | 9.1E-02 | NE | |
| lambdap74 | *S* | holin/anti-holin | 1.0E-05 | 4.6E-05 | E | E [114] |
| lambdap92 | *S'* | holin/anti-holin | 1.0E-05 | 1.7E-04 | E | E [114] |
| lambdap75 | *R* | endolysin | 6.0E-02 | 5.1E-02 | NE | E [115] |
| lambdap76 | *Rz* | I-spanin | 5.0E-04 | 1.5E-01 | E | E [116] |
| | *Rz1* | | | | NT | E [116] |
| lambdap77 | *bor* | serum resistance lipoprotein Bor | 7.9E-01 | 3.0E-01 | NE | NE [17] |
| lambdap78 | - | putative envelope protein | 1.9E+00 | 1.3E+00 | NE | NE [17] |
| lambdap79 | - | hypothetical protein | 3.0E-04 | 1.1E-04 | E | NE [17] |
| | | | | | | |
| ***Assay with pQ plasmid*** | | | | | | |
| lambdap62 | *ren* | protein ren | 1.20E-02 | 1.7E+00 | NE | NE [113] |
| lambdap63 | *ninB* | recombination protein NinB | 3.00E-01 | 9.1E-01 | NE | NE [17,65] |
| lambdap64 | *ninC* | | 1.00E+00 | 1.5E+00 | NE | NE [17,65] |
| | *ninD* | | | | | NE [17,65] |
| | *ninE* | | | | | NE [17,65] |
| lambdap67 | *ninF* | | 0.03 | 1.4E+00 | NE | NE [17,65] |
| lambdap68 | *ninG* | recombination protein NinG | 0.2 | 2.1E-01 | NE | NE |
| | *ninH* | | | | | NE [17,65,113] |
| lambdap70 | *ninI* | serine/threonine phosphatase | 1 | 1.5E+00 | NE | NE [17,65] |

E, essential; NE, nonessential; NT, not tested.

Interestingly, the CRISPRi-mediated knockdown of a cluster of delayed early genes (*ren*, *ninB/C/F/G/I*) in the $P_R$ transcriptional unit indicated that all were essential for plaque-formation, contradicting well-established literature [17,63–65]. This "*nin* region" lies between the essential DNA replication genes *O* and *P* and the *Q* gene, encoding the essential late transcription antiterminator. It is known that phages with a deletion of all the *nin* genes retain full plaque-forming ability [63–65]. The simplest interpretation for this discrepancy is that knockdowns in the *nin* region are polar on transcription of gene Q, the last gene in the

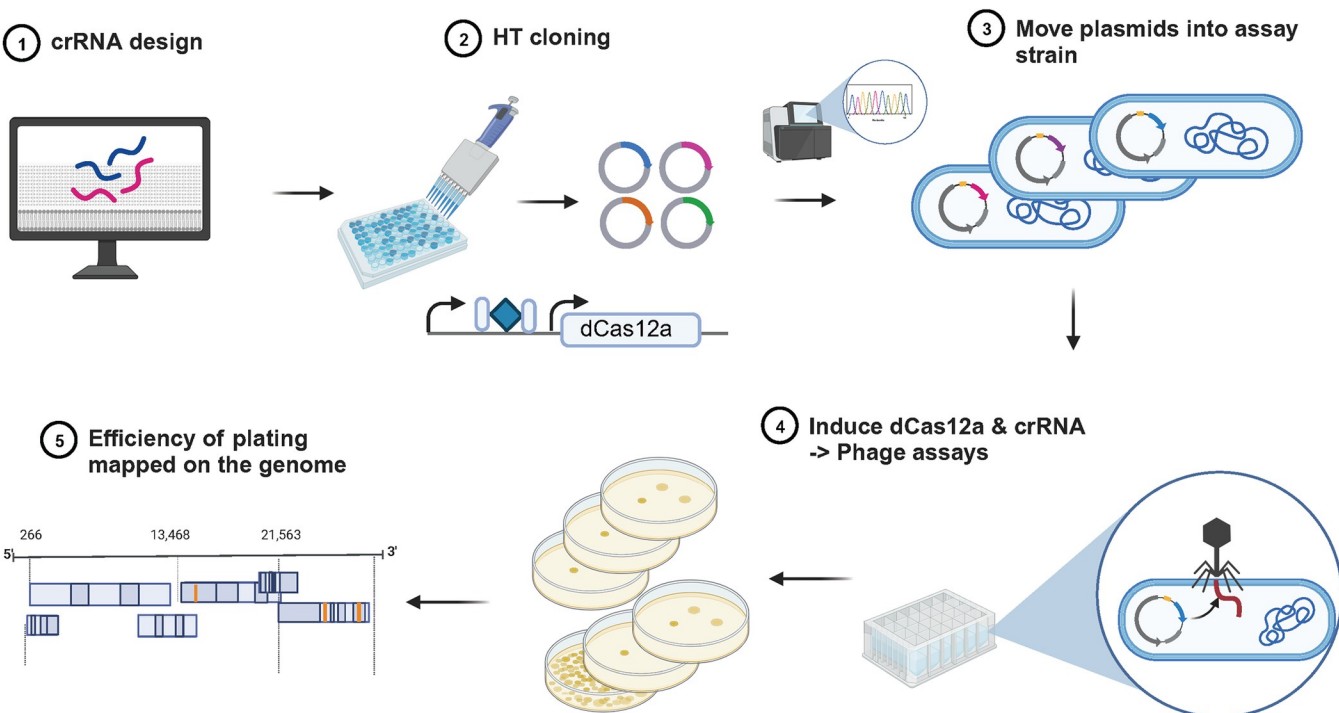

**Fig 2. Genome-wide CRISPRi design and assay format.** Schematic of steps involved in the arrayed CRISPRi knockdown experiments to assess gene essentiality in phage infectivity cycle. Created with BioRender.com

transcriptional unit. Polarity has been previously observed for CRISPRi knockdowns in a bacterial context, resulting in false positives in gene function assignments [66–70]. In lambda phage, all genes past *cro* are subject to N-mediated antitermination [17,71], and to our knowledge, CRISPRi knockdowns and polarity effects have not been tested with phage encoded antitermination systems. To determine whether the essential phenotype of the *nin* region genes in our assays is due to polarity on gene *Q*, we repeated the knockdown assays on an indicator strain that provides Q in *trans* from an inducible plasmid [72]. In these conditions, all 5 genes in the *nin* region targeted by CRISPRi were found to be nonessential, whereas providing Q had no effect on the essentiality of O and P (Figs 3 and S3). These results also conclude that dCas12a-mediated CRISPRi knockdown repression is insensitive to N-mediated antitermination. The Q protein is also an antiterminator and is required for expression of the 27 genes of the late transcript [17,71]. Although most of the genes of this transcript are known to be essential and score that way in our knockdown assays, two of the most promoter-proximal genes score as nonessential, including lambda *orf64* and, to a partial degree, *R*, which shows an intermediate plaque-forming defect. While *R* encodes the endolysin required for lysis, it is known to be produced in great excess, so a significant knockdown might still generate enough bacteriolytic activity to account for the intermediate plaque defect. *Orf64* is indicated to be nonessential [17], but it is unclear why the knockdown is not polar on the many essential genes downstream.

### Extending genome-wide CRISPRi assay to coliphage P1

We next extended the genome-wide CRISPRi knockdown assays to assess gene essentiality in coliphage P1. The 93-Kbp genome of P1 is composed of 117 genes, organized into 45 transcriptional units, with 8 involved in the lysis-lysogeny switch and plasmid prophage

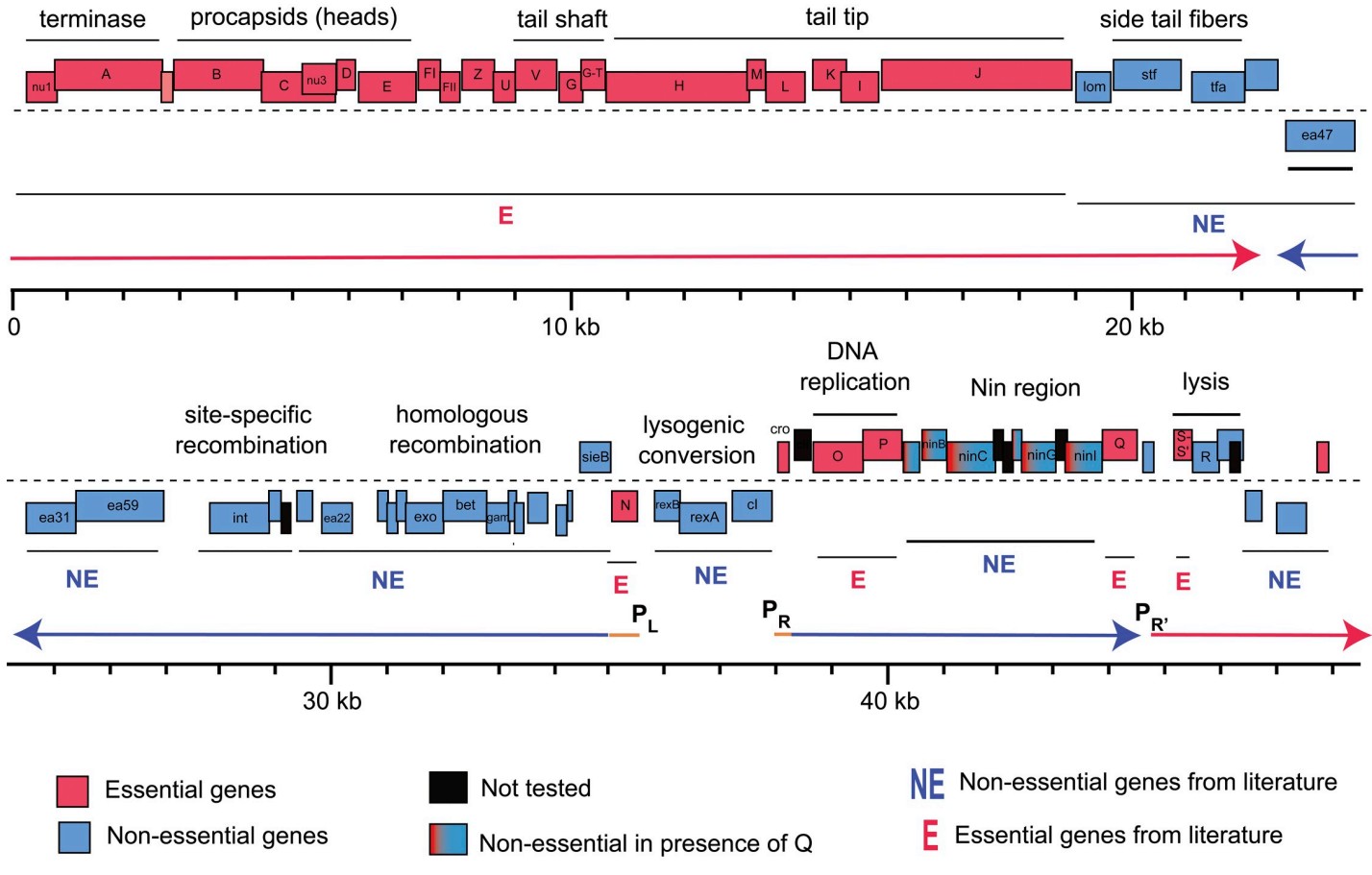

**Fig 3. Gene essentiality landscape of phage λ.** The genome-wide map of gene essentiality is shown by calculating the EOP as the ratio of plaques appearing on *E. coli* BW25113 lawn expressing crRNA targeting respective lambda phage genes to plaques appearing on BW25113 lawn expressing a nontargeting crRNA. The EOP estimations were done by carrying out biological replicates and depicted the average EOP of every gene on the lambda phage genome map (Methods). Transcripts mentioned in the main text (with promoters) are indicated as thick horizontal arrows: orange, immediate-early transcripts; purple, early transcripts; and red, late transcripts. The underlying data for this figure can be found in Table 1 and S1 Data.

maintenance, while 37 are involved in lytic development [52]. Despite its paradigm status, a large proportion of gene function assignments still awaits experimental verification [52,73]. Early gene expression and the lytic-lysogenic decision are controlled by the primary phage repressor C1, while Lpa (Late Promoter Activator) positively regulates late transcription. There are 11 late promoters, all of which have a conserved 9 bp inverted repeat that serves as the Lpa-binding site. Compared to lambda, there is no direct experimental evidence of a protein playing a role of antitermination in P1. Nevertheless, there are strong indications that P1 does encode antiterminators [52,74]. Among the 117 genes, 30 have been identified as essential for plaque formation by amber mutant and targeted deletion methods (Table 2 and Notes B in S1 Text). Experimental evidence for nonessentiality is available for 55 other genes, which makes P1 nearly as good for benchmarking the CRISPR knockdown strategy as lambda.

We designed individual crRNAs targeting 114 out of the 117 genes similar to lambda phage (above); the remaining 3 genes (*upfM*, *pdcA*, and *imcA*) were not tested due to lack of PAM sites. Using the same workflow described for lambda, we found 87 genes as nonessential and 27 genes as essential (Figs 4 and S4). Five known essential genes were missed by the knockdown screen: *mat*, *repL*, *25*, *26*, and *pmgR*. In addition, one gene, pmgN, was found to be

**Table 2. Gene essentiality mapping of Phage P1 genome.**

| locus_tag | gene | function [52] | EOP_average | SD | This work | Literature |
|---|---|---|---|---|---|---|
| P1_gp002 | cra | cre associated function | 6.6E-01 | 1.3E-01 | NE | NE [117,118] |
| P1_gp003 | cre | cyclization recombinase | 1.2E+00 | 8.1E-01 | NE | NE [117,118] |
| P1_gp004 | c8 | establishment of lysogeny | 1.5E+00 | 3.9E-01 | NE | |
| P1_gp005 | ref | recombination enhancement | 1.3E+00 | 3.5E-01 | NE | NE [118] |
| P1_gp006 | mat | maturation control | 2.8E-01 | 3.5E-03 | NE | E [119] |
| P1_gp007 | res | restriction component | 1.2E+00 | 3.5E-02 | NE | NE [120] |
| P1_gp008 | mod | modification component | 1.0E+00 | 3.2E-01 | NE | NE [120] |
| P1_gp009 | lxc | modulator of C1 action; | 7.0E-01 | 4.2E-01 | NE | NE [121] |
| P1_gp010 | ulx | enhances incorporation of darB | 6.4E-01 | 1.6E-01 | NE | NE [122] |
| P1_gp011 | darB | antirestriction | 5.8E-01 | 2.5E-01 | NE | NE [122] |
| P1_gp012 | prt | portal | <3.4E-6 | | E | E [52] |
| P1_gp013 | pro | head processing | <3.4E-6 | | E | E [52] |
| P1_gp115 | lydE | putative antiholin | 9.8E-01 | 8.8E-01 | NE | |
| P1_gp014 | lydD | putative holin | 1.1E+00 | 1.0E-01 | NE | |
| P1_gp015 | lyz | lysozyme | 5.0E-02 | 3.5E-03 | E | E [123] |
| P1_gp016 | ssb | single stranded DNA binding protein | 5.7E-01 | 2.0E-02 | NE | |
| P1_gp017 | isaA | IS1 insertion-associated gene | 8.6E-01 | 4.7E-01 | NE | |
| P1_gp018 | insB | IS1 transposition protein | 5.1E-01 | 1.6E-01 | NE | |
| P1_gp019 | insA | IS1 transposition protein | 1.0E+00 | 2.2E-01 | NE | |
| P1_gp020 | isaB | IS1 insertion-associated gene | 1.4E+00 | 2.2E-01 | NE | |
| P1_gp021 | hxr | possible repressor; homolog of Xre | 1.6E+00 | 1.2E+00 | NE | NE [122] |
| P1_gp022 | ddrB | antirestriction | 8.5E-01 | 1.3E-01 | NE | NE [122] |
| P1_gp116 | iddB | internal to ddrB | 5.4E-01 | 2.6E-01 | NE | NE [122] |
| P1_gp023 | ddrA | antirestriction | 9.8E-01 | 3.1E-01 | NE | NE [122] |
| P1_gp024 | darA | antirestriction | 6.5E-01 | 2.9E-01 | NE | NE [122] |
| P1_gp025 | hdf | antirestriction | 1.2E+00 | 8.4E-01 | NE | NE [122] |
| P1_gp026 | lydB | lysis determinant; prevents premature lysis | 2.3E-02 | 2.6E-02 | E | NE [123,124] |
| P1_gp027 | lydA | holin | 1.2E-01 | 2.4E-03 | NE | NE [123,124] |
| P1_gp028 | lydC | holin | 1.1E+00 | 1.2E-01 | NE | |
| P1_gp029 | cin | site-specific recombinase | 1.4E+00 | 9.5E-01 | NE | NE [125] |
| P1_gp001 | Sv prime | C-terminal moiety of tail fiber gpS | 5.3E-01 | 3.2E-01 | NE | NE [126] |
| P1_gp030 | U prime | structural protein gpU prime of tail fiber | 8.6E-01 | 8.0E-01 | NE | NE [126] |
| P1_gp031 | U | tail fiber structure or assembly | <5.2E-5 | | E | |
| P1_gp032 | S | tail fiber structure or assembly | <5.2E-5 | | E | |
| P1_gp033 | R | tail fiber structure or assembly | <5.2E-5 | | E | E [127] |
| P1_gp034 | 16 | baseplate or tail tube | <5.2E-5 | | E | E [128] |
| P1_gp035 | bplA | putative baseplate structure, may correspond to gene 3 | <2.9E-5 | | E | |
| P1_gp036 | pmgA | Putative morphogenetic function | <2.9E-5 | | E | E [73] |
| P1_gp037 | sit | putative tape measure protein | <2.9E-5 | | E | |
| P1_gp038 | pmgB | Putative morphogenetic function | <2.9E-5 | | E | E [73] |
| P1_gp039 | tub | tail tube | <2.9E-5 | | E | |
| P1_gp040 | pmgC | Putative morphogenetic function | <2.9E-5 | | E | E [73] |
| P1_gp041 | simC | superimmunity | 1.7E+00 | 6.9E-01 | NE | NE [129] |
| P1_gp042 | simB | superimmunity | 3.8E+00 | 8.4E-01 | NE | NE [129] |
| P1_gp043 | simA | superimmunity | 9.0E-01 | 4.9E-01 | NE | NE [129] |

*(Continued)*

**Table 2.** (Continued)

| locus_tag | gene | function [52] | EOP_average | SD | This work | Literature |
|---|---|---|---|---|---|---|
| P1_gr044 | c4 RNA | acts on icd and ant mRNA | 8.3E-01 | 9.4E-01 | NE | |
| P1_gp045 | icd | reversible inhibition of cell division | 6.0E-01 | 2.2E-01 | NE | NE [130] |
| P1_gp046 | ant1 | antagonizes C1 represssion1 | 2.0E+00 | 1.1E+00 | NE | NE [131] |
| P1_gp047 | ant2 | product antagonizes C1 | 2.3E+00 | 1.8E+00 | NE | NE [131] |
| P1_gp048 | ask | regulatory region of kilA gene; | 1.2E+00 | 4.9E-01 | NE | |
| P1_gp049 | kilA | product can kill host | 8.6E-01 | 8.0E-01 | NE | NE [131] |
| P1_gp050 | repL | initiates replication at oriL | 7.1E-01 | 3.0E-01 | NE | NE [131] |
| P1_gp051 | rlfA | possibly associated with lytic replication | 1.3E+00 | 6.1E-02 | NE | |
| P1_gp052 | rlfB | possibly associated with lytic replication | 8.8E-01 | 3.0E-02 | NE | |
| P1_gp053 | pmgF | putative morphogenetic function | 1.2E+00 | 1.0E-02 | NE | NE [73] |
| P1_gp054 | bplB | baseplate structure | <1.2E-7 | | E | |
| P1_gp055 | pmgG | putative morphogenetic function | <1.2E-7 | | E | E [73] |
| P1_gp056 | 21 | baseplate or tail tube | <1.2E-7 | | E | * [128] |
| P1_gp057 | 22 | tail sheath | <1.2E-7 | | E | * [128] |
| P1_gp058 | 23 | Major head protein | <1.2E-7 | | E | * [128] |
| P1_gp059 | parB | active partitioning of P1 plasmid during cell division | 1.3E+00 | 7.6E-01 | NE | |
| P1_gp060 | parA | active partitioning of P1 plasmid during cell division | 1.6E+00 | 1.5E+00 | NE | |
| P1_gp061 | repA | initiates replication from oriR; plasmid replication | 9.5E-01 | 6.4E-02 | NE | |
| P1_gp062 | upfA | | 1.2E+00 | 8.9E-01 | NE | NE [73] |
| P1_gp063 | mlp | membrane lipoprotein precursor | 2.4E-01 | 1.9E-01 | NE | |
| P1_gp064 | ppfA | possible periplasmic function | 1.3E+00 | 1.2E+00 | NE | |
| P1_gp065 | upfB | | 1.3E+00 | 1.0E+00 | NE | NE [73] |
| P1_gp066 | upfC | | 1.1E+00 | 6.8E-01 | NE | NE [73] |
| P1_gp067 | uhr | | 8.6E-01 | 1.9E-01 | NE | NE [73] |
| P1_gp068 | hrdC | hpothetical recombination associated protein of RdgC family | 8.7E-01 | 4.6E-01 | NE | |
| P1_gp069 | dmt-B | DNA methlytransferases; methlysates A at GATC | 1.4E+00 | 1.2E+00 | NE | NE [132] |
| P1_gp070 | dmt-A | | 1.4E+00 | 7.6E-03 | NE | |
| P1_gt071 | trnT | | 9.2E-01 | 4.0E-01 | NE | |
| P1_gp072 | plp | putative lipoprotein | 1.6E+00 | 8.2E-01 | NE | |
| P1_gp073 | upl | | 1.9E+00 | 5.9E-01 | NE | NE [73] |
| P1_gp074 | tciA | tellurite or colicin resistance or inhibition of cell division | 6.5E-01 | 6.5E-02 | NE | |
| P1_gp075 | tciB | tellurite or colicin resistance or inhibition of cell division | 9.9E-01 | 1.0E-01 | NE | |
| P1_gp076 | tciC | tellurite or colicin resistance or inhibition of cell division | 1.3E+00 | 4.4E-01 | NE | |
| P1_gt117 | trnI | | 1.4E+00 | 7.9E-01 | NE | |
| P1_gp077 | ban | dnaB homolog | 1.2E+00 | 4.8E-01 | NE | NE [133] |
| P1_gp078 | dbn | downstream of ban | 1.3E+00 | 1.3E-01 | NE | NE [73] |
| P1_gp079 | 5 | baseplate | <2.1E-6 | | E | * [128] |
| P1_gp080 | 6 | tail length | 8.3E-03 | 1.3E-03 | E | * [128] |
| P1_gp081 | 24 | baseplate or tail stability | 4.9E-03 | 5.0E-03 | E | * [128] |
| P1_gp082 | 7 | tail stability | <8.2E-6 | | E | * [128] |

(Continued)

**Table 2.** (Continued)

| locus_tag | gene | function [52] | EOP_average | SD | This work | Literature |
|-----------|------|---------------|-------------|-----|-----------|------------|
| P1_gp083 | 25 | tail stability | 1.1E-01 | 6.6E-02 | NE | * [128] |
| P1_gp084 | 26 | baseplate; | 1.2E-01 | 1.5E-01 | NE | * [128] |
| P1_gp085 | pmgL | putative morphogenetic function | 1.3E+00 | 9.4E-01 | NE | NE [73] |
| P1_gp086 | pmgM | putative morphogenetic function | 3.8E-01 | 2.1E-01 | NE | NE [73] |
| P1_gp087 | pmgN | putative morphogenetic function | 1.4E-02 | 1.5E-03 | E | NE [73] |
| P1_gp088 | pmgO | putative morphogenetic function | 1.1E+00 | 8.7E-01 | NE | NE [73] |
| P1_gp089 | pmgP | putative morphogenetic function | 7.2E-01 | 7.3E-02 | NE | NE [73] |
| P1_gp090 | ppp | protein phosphatase | 1.2E+00 | 8.0E-02 | NE | NE [73] |
| P1_gp091 | pmgQ | putative morphogenetic function | 1.4E+00 | 7.5E-01 | NE | NE [73] |
| P1_gp092 | pmgR | putative morphogenetic function | 9.4E-01 | 3.0E-01 | NE | E [73] |
| P1_gp093 | pmgS | putative morphogenetic function | 1.2E+00 | 3.2E-01 | NE | NE [73] |
| P1_gp094 | pap | acid phosphatase | 7.7E-01 | 2.3E-01 | NE | NE [73] |
| P1_gp095 | pmgT | putative morphogenetic function | 9.8E-01 | 9.7E-01 | NE | NE [73] |
| P1_gp096 | pmgU | putative morphogenetic function | 2.1E-01 | 6.4E-02 | NE | NE [73] |
| P1_gp097 | pmgV | putative morphogenetic function | 1.9E+00 | 2.0E+00 | NE | NE [73] |
| P1_gp098 | upfM | unknown protein function | | | NT | NE [73] |
| P1_gp099 | upfN | unknown protein function | 8.3E-01 | 2.4E-01 | NE | NE [73] |
| P1_gp100 | upfO | unknown protein function | 9.5E-01 | 6.4E-01 | NE | NE [73] |
| P1_gp101 | hot | DNA replication | 1.2E+00 | 1.6E+00 | NE | |
| P1_gp102 | lxr | LexA-regulated functions | 7.9E-01 | 7.7E-01 | NE | |
| P1_gp103 | humD | DNA repair | 5.4E-01 | 3.3E-01 | NE | |
| P1_gp104 | phd | anti-toxin of P1 toxin-antitoxin system | 5.5E-01 | 7.1E-02 | NE | |
| P1_gp105 | doc | toxin of P1 toxin-antitoxin system | 3.6E-01 | 1.2E-02 | NE | |
| P1_gp106 | pdcA | unknown protein function | | | NT | NE [73] |
| P1_gp107 | pdcB | unknown function | 6.8E-01 | 2.6E-01 | NE | NE [73] |
| P1_gp108 | lpa | late promoter activator | <1.2E-7 | | E | * [134] |
| P1_gp109 | pacA | DNA packaging | <1.2E-7 | | E | * [135] |
| P1_gp110 | pacB | DNA packaging | <1.2E-7 | | E | * [135] |
| P1_gp111 | c1 | lytic repressor | 1.4E+00 | 1.3E+00 | NE | |
| P1_gp112 | coi | C1 inactivator | 7.4E-01 | 8.4E-01 | NE | NE [136] |
| P1_gp113 | imcB | immunity function | 8.2E-01 | 7.3E-01 | NE | |
| P1_gp114 | imcA | immunity function | | | NT | |

E, essential; NE, nonessential; NT, not tested.

* amber mutant reported

essential, in contradiction with the recent deletion analysis survey [73]. From the perspective of identifying nonessential genes, 54 of the 55 genes for which there was some evidence of non-essential character were confirmed by the knockdown. In addition, the knockdown approach demonstrates nonessentiality for a further 33 genes. Taken together, 4 large segments comprising nearly 60 kb of the P1 genome are occupied by genes dispensable for lytic growth and thus available for specific engineering (Table 2).

## Downstream application of gene essentiality mapping

To demonstrate one downstream application of the knockdown approach to gene essentiality mapping, we sought to insert a unique DNA tag into both λ and P1 at a gene locus that we

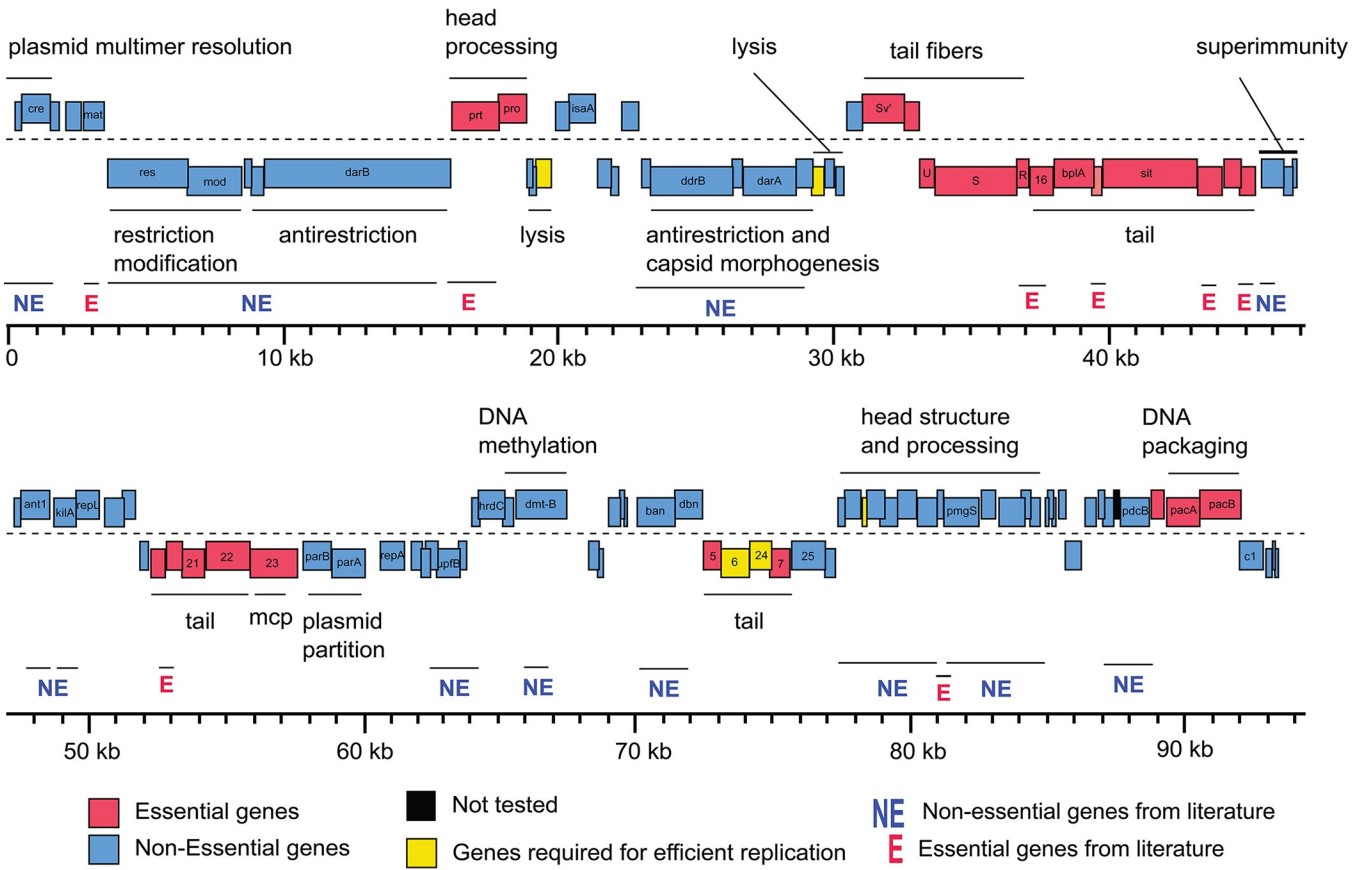

**Fig 4. Gene essentiality landscape of phage P1.** The genome-wide map of gene essentiality is shown by calculating the EOP as the ratio of plaques appearing on *E. coli* BW25113 lawn expressing crRNA targeting respective P1 phage genes to plaques appearing on BW25113 lawn expressing a nontargeting crRNA. The EOP estimations were done by carrying out biological replicates and depicted the average EOP of every gene on the P1 phage genome map (Methods). The underlying data for this figure can be found in Table 2 and S1 Data.

found to be dispensable. As DNA barcodes are heritable, they can be used for rapid identification of different phage samples by standardizing the workflow, assuming their insertion does not impact phage fitness. Such unique barcoding of different phages could enable quantitative tracking and measure of individual phage fitness in multiphage formulations in different applications.

As a proof of concept, we inserted a unique DNA barcode in genes *res* and *red*, of P1 and lambda, respectively. We used a homologous recombination approach followed by nuclease active Cas12a-based counter selection for barcoded phages in a 2-step process (Methods). Successful DNA barcode insertions into phage genomes were then confirmed by Sanger sequencing of the insertion locus. With these 2 *bc* (barcoded) constructs, we tested whether we could quantify different phage combinations. To do this, we mixed phage P1-bc and λ-bc in different ratios, incubated at room temperature for 30 minutes, and subjected them to Barseq PCR sequencing [75,76]. Our Barseq quantification method not only successfully quantified different ratios of barcoded phage P1 and lambda but also captured the differences in plaque-forming units/ml of individual phages to barcode abundance (Fig 5).

## Discussion

CRISPR-based technologies have revolutionized the functional genomics field [34]. CRISPRi, in particular, has emerged as a major technology for genome-wide mapping of essential and

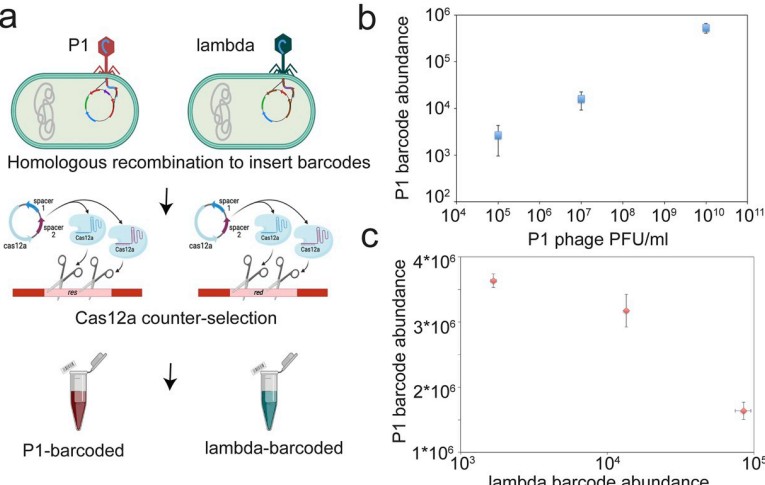

**Fig 5. Insertion and quantification of random DNA barcodes on a nonessential genomic location of lambda and P1vir phage.** (**a**) Schematic of phage engineering approach: Homologous recombination method was used to engineer phages with random barcodes at a nonessential genomic loci, and nuclease active Cas12a-based counterselection was used to enrich engineered phages. Schematic is shown for barcode insertion and counterselection for lambda phage at the *red* locus and P1 phage at *res* locus. Created with BioRender.com. (**b**) Barcode abundance of P1 phage against its PFU/ml estimations in triplicates. (**c**) Barcode abundance for both barcoded lambda and P1 phages, when mixed at different ratios. Estimations done in triplicates in a pool (Methods). The underlying data for this figure can be found in S1 Data.

nonessential genes in bacteria [34,35]. Here, we assessed the feasibility of using dCas12a system for performing a genome-wide survey of 2 paradigm phages, lambda and P1, using crRNAs designed to achieve gene-specific "knockdown." Results from our arrayed CRISPRi assays are consistent with known assignments of gene essentiality in both phages, provide novel insights, and present a genome-wide landscape of gene essentiality for phage P1 for the first time, to the best of our knowledge (Tables 1 and 2 and Figs 3 and 4). Lambda and P1 phages have quite distinct transcriptional organization, making CRISPRi differentially suited for probing stretches of nonessential genes in these phages (below). With an organized map of gene essentiality in hand, it is now possible to identify locations in these phage genomes wherein insertion of an exogenous "payload" are less likely to disrupt critical function, as well as longer regions that can be deleted or replaced with custom DNA. As a proof of principle, we demonstrate this by inserting a DNA barcode into the lambda and P1 genome at an inessential loci that provides the ability to track and quantify distinct phages in a mixed phage formulation. Finally, this study uncovers the polar effect of CRISPRi in phages. We recommend using CRISPRi for mapping nonessential regions while caution towards interpreting essential gene assignments when applied to less studied phages where transcripts have not yet been mapped. We discuss these insights below.

Overall, the genome-wide CRISPRi assay results demonstrated dCas12a was effective; that is, nearly every nonessential lambda gene knockdown was scored correctly, and essential lambda genes were scored as essential, based on reduction of plating efficiency by 3 powers of 10 or more in the presence of dCas12a and the cognate crRNA (Fig 3 and Table 1). However, a cluster of delayed early genes in the *nin* region of the $P_R$ transcript of lambda were scored as essential despite unambiguous evidence that this entire region can be deleted without impairing the plaque-forming ability of the phage [63–65]. Because of its DNA-binding function, the bound dCas12a/crRNA complex is necessarily a roadblock that would be polar on all

downstream genes, as confirmed experimentally for the knockdowns of *lacZ* in the *lacZYA* operon of *E. coli* [38]. The reason for this polarity is because this cluster of nonessential *nin* genes is upstream of gene *Q*, which encodes the late-gene activator required for late-gene expression. Thus, roadblocks in the *nin* genes should be polar on gene Q. Accordingly, when we supplied Q in *trans*, the *nin* genes all scored properly as nonessential (Fig 3). Unfortunately, the same rationale applies to the other genes served by P$_R$ (Fig 3). Thus, knockdowns in *cro*, *O*, and *P* are also polar on Q. With *Q* added in *trans*, all 3 upstream genes read out as essential but only the gene *P* result is confirming, since the *cro* and *O* knockdowns should be polar on essential gene *P*. The situation is better for the P$_L$ transcriptional unit because the only essential gene is the first one, *N*. Thus, for P$_L$, all 19 genes that were tested are scored correctly, as nonessential.

Similar challenges for CRISPRi essentiality determination are noted for the late genes, expressed from P$_{R'}$ in a 27-kb mRNA (Fig 3). Twenty-one genes from *nu1* through *J* read out correctly as essential, but since the first 20 knockdowns should be polar on J, nothing can be concluded for their essentiality based on CRISPRi results. Moreover, the results for the upstream genes *orf64* and *R* are confounding. From the same perspective as used on the P$_R$ transcript, knockdown roadblocks in all the upstream genes in this transcriptional unit should be read out as essential. This was observed for gene *p79* (which is nonessential, but essential in our assay), but it was not observed for the knockdowns of *orf64* and *R*. In CRISPRi studies on bacterial genomes [66–70,77], similar polarity issues have been noted, and contradictions have been explained by invoking the presence of cryptic promoters downstream of the roadblock site [69]. For a phage like lambda, where transcriptional organization has been unambiguously established by rigorous genetics and molecular approaches (though new technologies are providing new information [78]), these arguments may not hold. The simplest possibility is that there are large variations in the effectiveness of each roadblock [79], despite the perfect match of 28 nucleotides in each crRNA and, in each case, a TTTV PAM sequence. Hence, in the absence of data assessing the level of readthrough in the *orf64* and *R* roadblocks, useful interpretation of the P$_{R'}$ results is not practical. An intriguing possibility is that Q-mediated antitermination may play a role in readthrough of these CRISPRi roadblocks. It is widely unappreciated that for all the well-studied phages, late gene expression is always under positive control, either by an antiterminator like lambda Q and the P$_{R'}$ promoter or by a transcription factor like Lpa and the 11 late promoters of P1 [17,52]. It would be interesting to determine quantitatively how such positive control factors affect the efficacy of Cas12a in CRISPR defense and dCas12a in roadblock knockdowns, with an eye towards possible evolutionary interactions. In any case, the results from knockdowns in the 3 major transcripts of lambda show that only *N*, *P*, *Q*, and *J* can be confidently established as essential genes. Thus, as noted earlier [66], the nature of CRISPRi roadblock polarity means that essentiality can only be assigned for the last required gene on a transcript. The 2 major lessons from our work on lambda are, first, CRISPRi polarity could assign false positive gene essentiality and therefore recommend caution when applied to less studied phages; and, second, CRISPRi based on DNA roadblocks is of limited utility for analysis of phages that, like lambda, feature long polycistronic transcriptional units. However, for the more utilitarian goal of identifying significant swaths of the phage genome that could be considered "nonessential" and thus available for engineering, this approach still has high value. All of the 14-kb P$_L$ transcript beyond N, comprising 15 genes, score unambiguously as nonessential.

Among phage genomes, lambda is arguably the best characterized transcriptional system because of its simplicity, with only 3 promoters involved in lytic development. P1 stands in stark contrast, with at least 45 transcriptional units, including 15 monocistronic units, and several genes served by both early and late promoters. In general, similar results were obtained

from the genome-wide knockdown approach as with lambda (Fig 4 and Table 2). Of the 31 genes assigned essential character in the extensive P1 literature, all but 5 were detected by the knockdown screen. However, consideration of the transcript structures and gene positions reveals that of the 26 genes that read out as essential, 18 are located upstream of a gene known to be essential, and, thus, the knockdown readout is uninformative. Moreover, as in the case for the promoter-proximal genes in the lambda late transcriptional unit, P1 has a confounding transcript. Genes *25* and *26*, which were discovered as amber mutants and thus must be considered as known essentials, both score as nonessential genes in our assays. This constitutes a double contradiction, not only in the failure to detect essential character but also not exhibiting polarity on the cluster of genes downstream (genes *7*, *24*, *6*, and *5*) that correctly read out as required cistrons. The simplest notion is that for some reason, neither the *25* nor *26* roadblocks are effective. Quantitative assessment of roadblock readthrough is beyond the scope of this initial validation screen, but it would be useful to determine the level of blockage and readthrough throughout the lambda and P1 libraries (as recently reported for *E. coli* [79]). This is especially true since the 2 confounding cases (genes *orf64* and *R* in lambda; genes *25* and *26* in P1) are at the 5′ end of a polycistronic transcriptional unit. Unlike other CRISPRi systems, the dCas12a roadblocks are reported to be independent of promoter-proximity, but that lesson has only been addressed within the *lacZYA* cistron [38], and not for very long transcripts or for transcriptional units under the positive control of an antiterminator.

Because of tightly overlapped and transcriptionally linked genetic elements in phages, such polarity effects may be difficult to overcome using CRISPRi. The catalytically inactive version of recently reported RNA-targeting Cas13 system might solve some of the polarity effect issues associated with DNA-targeting Cas systems by modulating translation of single genes encoded within operons [29,31]. In addition, the absence of PAM requirements for Cas13 targeting and its broad-spectrum phage targeting capability may enable designing multiple crRNA targeting the same genomic locus, to quickly and comprehensively map gene essentiality landscape in diverse phages [29]. Nevertheless, in contrast to classical genetic methods such as recombineering, that require cumbersome cloning of long homology arm pairs followed by plaque screening to identify edited phages that exist at low abundance relative to wild-type, arrayed CRISPRi assay as presented here offers a simpler and economical approach that only requires cloning a set of short crRNA sequences. By using pooled crRNAs, it may be possible to extend the CRISPRi technology to carry out pooled fitness assays and identify phage genes important in the phage life cycle in a single rapid assay. While this manuscript was under review, successful implementation of dCas13 based genome-wide pooled CRISPRi screen was reported for diverse phages [80] and point to a rich future of diverse functional genomics tools to study phage biology.

Even though the gene essentiality mapping results are dependent on the experimental settings and conditions used in the assay systems, they do open up interesting questions and avenues to assess the role of nonessential and accessory genes in phage development and infection pathways [15,17,18]. By adopting high-throughput CRISPRi assays to map phage gene essentiality in different conditions [81], it may be possible to study the role of such conditional gene essentiality in phage infection. Furthermore, the simple multiplexability of dCas12a crRNAs (for example, dual crRNAs targeting 2 genes) could enable rapid, systematic investigation of synthetic lethal phage gene pairs. Extending such studies to non-model, non-dsDNA phages may further provide us with deeper information needed to study genomic architecture and phage engineering applications. Considering that the different CRISPR-based tools have been successfully applied to multitudes of microbial species [34,82] and have been used to engineer diverse phages, we expect CRISPRi technology to serve as a powerful approach to rapidly identify nonessential and accessory genes and pathways in phage infection cycles.

## Methods and materials

### Bacterial strains and phages

The bacterial strains and phages used in this study are listed in S1 Table. The oligonucleotides used in this study are listed in S2 Table. All enzymes were obtained from New England Biolabs (NEB), and oligonucleotides were received from Integrated DNA Technologies (IDT). Unless noted, all strains were grown in LB supplemented with appropriate antibiotics at 37˚C in the Multitron shaker. All bacterial strains were stored at −80˚C for long-term storage in 15% sterile glycerol (Sigma). The genotype of *E. coli* strains used in the assays include BW25113 (K-12 *lacI*+*rrnBT*14 Δ(*araB–D*)567 Δ(*rhaD–B*)568 Δ*lacZ4787*(::*rrnB*-3) *hsdR*514 *rph*-1); MG1655 (F-lambda- *ilvG*- *rfb*-50 *rph*-1) and *E. coli* C3000 (ATCC15597).

*E. coli* strains were cultured in LB (Lennox) [10 g/L Tryptone, 5 g/L NaCl, 5 g/L yeast extract] or LB agar [LB (Lennox) with 1.5% Bacto agar]] at 37˚C. *E. coli* strains transformed with plasmids were selected in the presence of 100 μg/mL ampicillin (LB Amp) or 30 μg/mL chloramphenicol (LB cam). Phages were plated using 0.5% top agar [10 g/L Tryptone, 10 g/L NaCl, 5 g/L Bacto agar]. Before plating, 5 mM CaCl2 and 5 mM MgSO4 were added to top agar aliquots.

The phages (lytic phages, λcI857 and P1vir) used in this study were prepared by the confluent plate lysis method using LB bottom plates and 0.5% top agar [83]. Phages were harvested in SM buffer (Teknova), filter sterilized, and stored at 4˚C. Plaque assays were performed using spot titration method [83].

### Design and construction of spacer duplex

Cas12a recognizes TTTV as the PAM site [53]. For each target gene, PAM sites for Cas12a were identified to serve as toe-holds for the crRNAs. As any genes could have an alternative start site, the PAM sites nearby the annotated start codon of the gene were avoided. To avoid end effects, and based on prior experience in bacterial CRISPRi [70], PAM sites were prioritized if they occurred after 20% of the gene length (so that the dCas12a complex would bind to approximately on the 1/5th position of the gene). The 28-bp nucleotide sequences immediately downstream of the PAM site in the coding strand were selected as the protospacer region. The forward oligo was designed by adding sequences "AGAT" to the 5′ region of the protospacer sequence and sequence "G" to the 3′ region of the protospacer sequence to make the ends of oligos Golden Gate cloning compatible. The reverse oligo was designed by reverse complementing the protospacer sequence from the coding strand and adding sequences "GAAAC" to the 5′ end. Custom python scripts (https://github.com/NickNolan/phage-crispri) were designed for identifying the protospacer regions and respective oligonucleotides.

We processed oligonucleotides by carrying out 5′ phosphorylation and annealing of complementary oligonucleotides in a single tube reaction. The published sequences for phages P1 (NCBI Reference Sequence: NC_005856.1) and λ (NCBI Reference Sequence: NC_001416.1) were used as reference sequences to generate oligos. Each 5 μL reaction comprised 0.5 μL each of the forward and reverse oligonucleotide pair (100 μM stock), 0.5 μL of 10× T4 DNA Ligase Reaction Buffer (NEB), 0.5 μL T4 Polynucleotide Kinase (NEB). The reaction was carried out in a thermocycler as follows: 37˚C for 30 minutes, 95˚C for 5 minutes, followed by gradient decrease of temperature from 95˚C to 25˚C (0.5˚C every 6 seconds for 140 cycles). To make a working stock of the spacer duplex, the reaction mix was diluted to a final volume of 100 μL by adding milliQ water.

### Plasmid construction

The plasmid collection used in this study is listed in S3 Table. All plasmid manipulations were performed using standard molecular biology techniques. The plasmid system encoding

nuclease active LbCas12a has been described previously [53]. In brief, LbCas12a is cloned under aTc-inducible Tet promoter, whereas the CRISPR arrays are constitutively transcribed from a strong, synthetic promoter proD [53]. For CRISPRi, catalytically deactivated LbCas12a (dLbCas12a) lacking endonuclease activity was generated by the mutating nuclease domain of LbCas12a. For each CRISPRi plasmid, a spacer targeting a specific phage gene was cloned into the CRISPR array using Golden Gate assembly [84]. Each 5 µL of the reaction contained 0.5 µL of ATP (NEB), 0.5 µL DTT (1 mM final concentration), 0.5 µL 10× CutSmart Buffer (NEB), 0.375 µL BbsI (NEB), 0.125 µL T4 Ligase (NEB), 20 fmol CRISPRi plasmid, and 100 fmol spacer duplex (0.2 µL of the working stock of the spacer duplex). The reaction was cycled between 37°C and 20°C for 5 minutes each at each temperature for 30 cycles and heat inactivated at 80°C for 20 minutes. This same method was followed to clone the spacer duplex into the plasmid encoding nuclease active version of LbCas12a.

For inserting a random DNA barcode into a nonessential region of phage (*res* and *cra-darB* region in P1 phage while *red* gene in lambda; S3 Table), a recombination template was constructed on pBAD24 vector backbone [85]. A synthetic dsDNA was obtained from IDT as a gBlock gene fragment that comprised 2 homology arms, each of 100-bp homology to the non-essential region of the phage genome [86]. In between the two 100-bp homology arms, a random 20-bp DNA barcode flanked by 2 primer-binding regions was inserted so that the barcoded phage genome could be assayed by high-throughput DNA barcode sequencing (Bar-Seq) technology [75]. The gBlock fragment was PCR amplified and cloned into a PCR-amplified pBAD24 backbone using Gibson assembly [87].

The Golden Gate or Gibson assembly mixture was transformed into competent *E. coli* 5-alpha cells (NEB) following manufacturer's recommendations and selected by plating on LB in the presence of appropriate antibiotics. Successful insertion into the plasmid backbone was verified by Sanger sequencing (UC Berkeley DNA Sequencing Facility or Elim Biopharmaceuticals). These pBAD24-derived plasmids would serve as recombination templates.

## CRISPRi assays for mapping phage gene essentiality

For CRISPRi knockdown assays, each variant of the CRISPRi plasmid was transformed into *E. coli* str. BW25113 using standard method [88] and selected on independent LB cam plates. An overnight culture of the transformed strain was used to prepare a lawn on LB cam supplemented with 2 nM or 4 nM aTc for induction of the dCas12a. Phages were serially diluted 10-fold, and 2 µL of each dilution was plated on a lawn of bacterial host. The number of plaques was quantified after overnight incubation at 37°C. The EOP was calculated as the ratio of plaques appearing on BW25113 lawn expressing crRNA targeting respective phage genes to plaques appearing on BW25113 lawn expressing nontargeting crRNA. The nontargeting crRNA targets P1 phage gene *23* in lambda CRISPRi assays while it targets lambda phage gene *E* in P1 CRISPRi assays. The complete compendium of EOP for each CRISPRi knockdown assay for lambda and phage P1 is listed in Tables 1 and 2, respectively.

To assess the essentiality of the Nin region in λcI857, we transformed all *nin* targeting CRISPRi plasmids into *E. coli* str. BW25113, carrying a pQ plasmid system [72], and carried out CRISPRi knockdown assays as described above (S3 Fig). The plasmid pQ, a low-copy plasmid carrying Q, encodes the λ late gene activator under control of a *lac/ara* hybrid promoter, which is inducible with IPTG and arabinose.

To determine the conditional essentiality of *ral*, we transformed the *ral* targeting CRISPRi plasmid into *E. coli* MG1655 and C3000 strains and carried out CRISPRi knockdown assays in the presence of lambda phage as described above. Both *E. coli* MG1655 and C3000 strains encode an active type I restriction system. The plaque-forming efficiency was compared

between *E. coli* lawn expressing crRNA targeting *ral* and *P* genes and with plaques appearing on *E. coli* lawn expressing nontargeting crRNA.

## Engineering DNA barcoded phages

For inserting the DNA barcode into the phage genome, pBAD24-derived plasmid (S3 Table) was transformed into *E. coli* str. BW25113 using a 1-step transformation method [88]. Phage stock was appropriately diluted and plated on the lawn of the transformed BW25113 host using full-plate titration method [83]. Individual plaques were picked from the lawn, and the insertion of the DNA barcode was verified by PCR amplifying the junction and Sanger sequencing. The phages obtained from each plaque had a mixed population of unmodified and recombinant phage. This mixed population of phages were further enriched by confluent lysis plating method, and the wt phage in each plaque was counterselected by plating the mixture phage on the lawn of BW25113 host expressing nuclease active Cas12a target the nonessential region of the phage [89].

## Barseq assays using DNA barcoded phages

To demonstrate the utility of barcoded phages, we mixed uniquely barcoded P1 and lambda phage lysates in different ratios, in triplicates. To benchmark the barcoded phage quantification with a set of internal controls, we spiked 4 uniquely barcoded *E. coli* genome preparation into each of the Barseq samples. For performing Barseq PCR reactions, we used phage lysates as templates mixed with *E. coli* genome preparations. BarSeq PCR in a 50-μl total volume consisted of 20 μmol of each primer. We used an equimolar mixture of BarSeq_P2 primers along with new Barseq3_P1 primers as detailed earlier [75,90]. Briefly, the BarSeq_P2 primer contains the tag that is used for demultiplexing by Illumina software, and the new Barseq3_P1 primer contains an additional sequence to verify that it came from the expected sample (as described earlier) [90]. All experiments were done on the same day and sequenced on the same lane. Equal volumes (5 μl) of the individual BarSeq PCRs were pooled, and 50 μl of the pooled PCR product was purified with the DNA Clean and Concentrator kit (Zymo Research). The final BarSeq library was eluted in 40 μl water. The BarSeq libraries were sequenced on Illumina HiSeq4000 instrument with 50 SE runs. We used in-house Barseq PCR processing code for estimating DNA barcodes in samples [75].

## Supporting information

**S1 Fig. Lambda phage genome CRISPRi oligo designs.** Blue represents genes, red represents primers, and black is the full genome. Outside is on the positive strand, where inside is negative.
(TIF)

**S2 Fig. Conditional essentiality of lambda *ral* in presence of an active type I restriction-modification system encoded by *hsdR*-*hsdM*-*hsdS* genes.** (**a**) EOP experiments with crRNA targeting *ral* in *E. coli* BW25113 (methods). (**b**) EOP experiments with crRNA targeting *ral* in *E. coli* MG1655 that has an active type 1 restriction modification system. (**c**) EOP experiments with crRNA targeting *ral* in *E. coli* C3000 that has an active type 1 restriction modification system. For comparison, phage plaques appearing on *E. coli* lawn expressing a crRNA targeting essential gene *P* and nontargeting crRNA (targets P1 phage *mcp*) as a control are shown for lambda phage (Ctrl).
(TIF)

**S3 Fig. CRISPRi of Nin region without and with plasmid expression gene Q: EOP experiments for assessing CRISPRi polarity effect on Nin region.** (**A**) EOP for CRISPRi assay for each gene shown (each strain with individual CRISPRi plasmid (Methods). (**B**) EOP experiments in presence of plasmids pQ and CRISPRi targeting each gene in Nin region context. We used the BW25113 strain with a crRNA vector control (DP51, crRNA targets P1 phage *mcp*) for estimating EOP.
(TIF)

**S4 Fig. P1 phage genome, CRISPRi oligo designs.** Blue represents genes, red represents primers, and black is the full genome. Outside is on the positive strand, where inside is negative.
(TIF)

**S1 Table. List of bacterial strains and phages.**
(XLSX)

**S2 Table. List of primers used in this work.**
(XLSX)

**S3 Table. List of plasmids used in this work.**
(XLSX)

**S1 Text. Detailed discussion on phage CRISPRi results.**
(DOCX)

**S1 Data. The underlying data for all figures.**
(XLSX)

## Acknowledgments

The authors thank Dr. Jennifer Doudna (Innovative Genomics Institute, UC Berkeley) for sharing reagents and guidance. The authors also thank Dr. Benjamin Adler (Innovative Genomics Institute, UC Berkeley) for comments on the early draft of this manuscript.

## Author Contributions

**Conceptualization:** Denish Piya, Vivek K. Mutalik.

**Data curation:** Denish Piya, Ry Young, Adam P. Arkin, Vivek K. Mutalik.

**Formal analysis:** Denish Piya, Vivek K. Mutalik.

**Funding acquisition:** Adam P. Arkin, Vivek K. Mutalik.

**Investigation:** Denish Piya, Nicholas Nolan, Madeline L. Moore, Luis A. Ramirez Hernandez, Vivek K. Mutalik.

**Methodology:** Denish Piya, Nicholas Nolan, Luis A. Ramirez Hernandez, Brady F. Cress, Ry Young, Vivek K. Mutalik.

**Project administration:** Vivek K. Mutalik.

**Resources:** Brady F. Cress, Ry Young, Adam P. Arkin, Vivek K. Mutalik.

**Software:** Nicholas Nolan, Vivek K. Mutalik.

**Supervision:** Adam P. Arkin, Vivek K. Mutalik.

**Validation:** Denish Piya, Vivek K. Mutalik.

**Visualization:** Vivek K. Mutalik.

**Writing – original draft:** Denish Piya, Vivek K. Mutalik.

**Writing – review & editing:** Brady F. Cress, Ry Young, Adam P. Arkin, Vivek K. Mutalik.

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

26 / 26