## [Editor Report · Decision Letter 0]

2 Jun 2023

Dear Dr. Mutalik, 

Thank you for submitting your manuscript entitled "Genome-wide CRISPRi knockdown to map gene essentiality landscape in coliphages λ and P1" for consideration as a Research Article by PLOS Biology.

Your manuscript has now been evaluated by the PLOS Biology editorial staff, as well as by an academic editor with relevant expertise, and I am writing to let you know that we would like to send your submission out for external peer review as a Methods and Resources article.

Methods and Resources Articles describe technical innovations, including novel approaches to a previously inaccessible biological innovation, or substantial improvements over previously established methods. The reported method should be thoroughly validated, and while presenting new biological insights is encouraged, this is not a requirement for consideration. Resources consist of data sets or other significant scientific resources that are of general interest and provide exceptionally value for the community that could spur future research.

Once your full submission is complete, your paper will undergo a series of checks in preparation for peer review. After your manuscript has passed the checks it will be sent out for review. To provide the metadata for your submission, please Login to Editorial Manager (https://www.editorialmanager.com/pbiology) within two working days, i.e. by Jun 04 2023 11:59PM.

Kind regards,

Paula

---

Senior Editor

PLOS Biology

---

## [Decision Letter · Decision Letter 1]

7 Jul 2023

Dear Dr. Mutalik,

Thank you for your patience while your manuscript "Genome-wide CRISPRi knockdown to map gene essentiality landscape in coliphages λ and P1" was peer-reviewed at PLOS Biology. It has now been evaluated by the PLOS Biology editors, an Academic Editor with relevant expertise, and by several independent reviewers. 

In light of the reviews, which you will find at the end of this email, we would like to invite you to revise the work to thoroughly address the reviewers' reports.

As you will see below, reviewers #1 and #2 agree that the method will be very useful for the community, but all the reviewers find issues that will need to be solved before further consideration. Reviewer #3 asks for mechanistic insights, but these are not needed for Methods and Resources manuscripts. We think that you should address all the issues raised by reviewers #1 and #2 and especially the notion of whether this is broadly applicable to other phages. 

Given the extent of revision needed, we cannot make a decision about publication until we have seen the revised manuscript and your response to the reviewers' comments. Your revised manuscript is likely to be sent for further evaluation by all or a subset of the reviewers.

**IMPORTANT - SUBMITTING YOUR REVISION**

*Re-submission Checklist*

*Published Peer Review*

*PLOS Data Policy*

*Blot and Gel Data Policy*

Sincerely,

Paula

---

Senior Editor

PLOS Biology

REVIEWS:

Reviewer #1: Phage therapy.

Reviewer #2: Phage biology.

Reviewer #3: Phages and CRISPR.

Reviewer #1: This paper by Piya et al. primarily describes the deployment of CRISPRi via dCas12a to globally assess gene essentiality within two model bacteriophage. Using their identification of non-essential regions of the phage genomes in a practical example they then place genetic barcodes into phage lambda and P1 genomes and show how these can be used to enumerate and track phage populations within a mixed culture.

Overall, the paper is very clearly written and the logic and reasoning behind the experiments is well explained. The work is important and would be expected to be broadly applicable across at least the dsDNA phages, and likely ssDNA phages as well (although no example is given) that infect E. coli and closely related species. The plasmids as they are currently configured may not function in more distant host groups, but the principle outlined in this work should still hold if porting over to different bacterial families. The materials in the work are very well described and using the plasmids with different CRISPRi oligos would be straightforward. If the authors have not already, I encourage them to deposit the plasmids with Addgene so they are easily accessible.

I have a few comments:

Figures: I'm not sure if this is due to the journal's reformatting of PDFs, but most of the figures show as fuzzy and some text is illegible (e.g. Fig 1)

Both phage lambda and P1 are temperate phage and I was wondering as I read the paper how you were dealing with that possibility. I see it explained in the Materials and Methods, but it would be good to mention in the main text that variants being used in the work are lysogen-deficient.

The non-targeting crRNA was not fully described in the text. For example, what does it target? Is it similar GC-content to other oligos used? 

The pros and cons of using dCas12a versus dCas9 are not discussed sufficiently. Why was dCas12a chosen? Would you expect similar results with dCas9? Why or why not?

Amber mutants are treated as "ground truth" in this work, but have the authors thought about whether some amber mutants may also be polar mutations? Translation is thought to protect transcripts from degradation and if translation is stopped over a large section of mRNA this may destabilize the transcript causing a polar effect. One older article seems to show this may be possible (https://doi.org/10.1016/0022-2836(68)90144-7) although the phage is very different that lambda or P1.

Supplementary is misspelled in several different parts of the manuscript

Figure 4: the color choice for essential (red) and intermediate phenotype (purple) is not discernible in the figures. Need to change colors so they have more contrast. It would also be very helpful to harmonize the colors between Fig 3, 4 and Tables 1 and 2.

Table 2: The EOP average column has a large number of instances of "essential" instead of a quantitative measure, which is confusing. 

The "Downstream application" section is not introduced with sufficient detail to really understand what is going on with the experiments. For example, how long are the phage cultured before measuring? This is pertinent because small fitness defects would be expected to have a detectable effect over large numbers of generations. Fig 5C it is impossible to differentiate the symbols.

Reviewer #2: The manuscript by Piya et al. from the Arkin and Mutalik labs reports the application of CRISPRi to bacteriophages using a setup based on catalytically inactive Cas12a. In difference to previous work using similar technology, the authors target (almost) every gene in two phages and thus report on the first true genome-wide CRISPRi screens in bacterial viruses. This is certainly a breakthrough for the field, but in the current form constitutes rather quantitative than qualitative progress compared to previous work. The paper is very well written and structured comprehensibly. The methodology is described very nicely with considerable detail which would facilitate wide application of the authors' technology.

The introduction appropriately introduces the great need for genome-wide screens in bacteriophages to probe the biological functions of the vast amount of phage genes of unknown function. However, it is not optimal that CRISPR-Cas12a and its use for CRISPRi are just dropped onto the non-expert reader without further explanations. I would therefore suggest to (at least very shortly) introduce the key differences between CRISPR-Cas9 (that many readers have likely heard about) and CRISPR-Cas12a to explain why the authors chose the latter and not the former, even though CRISPR-Cas9 is by far most often used for microbial CRISPRi. Furthermore, I feel that previous use of genome-wide CRISPRi to study viral biology needs to be referenced (Hein and Weissman, Nature Biotechnology (2021)) and explained with view to the question why the same approach could not be simply applied to bacteriophages.

The results section begins with a nicely comprehensible and detailed description of how the authors set up CRISPRi using dCas12a. The authors' validation of the system's basic functionality to distinguish between essential and non-essential genes is very convincing. However, in my view the quantification of knockdown phenotypes in figure 1 is not optimal, though I acknowledge that this is a never-ending discussion in the field. The authors write about a 10'5- to 10'6fold reduction in plating efficiency (line 126). However, there is no plaque formation visible for the knockdowns of essential genes, only "lysis from without". I therefore find it misleading to talk about a reduction in plating because no plating (i.e., plaque formation) can be seen.

Subsequently, the authors use their CRISPRi approach to query (near-)genome-wide gene essentiality in phages lambda and P1 gene by gene. These phages were chosen because generations of researchers have already generated a considerable body of work on gene essentiality for these phages which provides a very convincing benchmark for the authors' technology. The results are indeed largely in line with expectations from previous literature. However, the authors also observe both polar effects and consequences of incomplete knockdowns. These are known downsides of CRISPRi per se and thus do not constitute a specific problem of the authors' approach, even though they later discuss (but do not try) overcoming at least some of the polar effects by using catalytically inactive Cas13a instead of Cas12a.

At the end, the authors inserted barcodes at different loci of their two model phages that their knockdown approach had identified or validated as being not essential. As expected, these barcodes can be used to quantify the abundance of different phage mutants in pools using BarSeq. However, in the current form these experiments seem added onto the manuscript without any real benefit. Different non-essential regions that could be targeted by barcode insertion have already been known before and a small barcode is hardly a difficult payload to insert, so this does not provide a significant advance and has otherwise nothing to do with the methodology that the authors present. In my view this section should either be removed or expanded in a way that connects it better to the rest of the work (e.g., by directly performing phenotypic selection experiments with pooled libraries of barcoded sgRNAs, if the authors like to use BarSeq for an approach similar to CRISPRi-Seq).

The discussion section appropriately places the authors' work into context of the literature and the research problem as defined in the introduction section. Subsequently, the authors discuss different shortcomings of their approach and possible solutions or applications. These parts were interesting but - because they paraphrased a lot of content from the results section - could be easily streamlined and shortened. The authors have included very nice supplementary notes with additional details that are of interest primarily for phage aficionados (greatly appreciated).

In my view this work definitely establishes genome-wide CRISPRi screens for bacteriophages, even though most of the technology has already been around and was used previously for phages (albeit certainly not at the same scale). Furthermore, even (kind-of) CRISPRi-Seq in a virus has already been reported previously for human cytomegalovirus (Hein and Weissmann, Nature Biotechnology 2021 (not cited)). I therefore feel that the authors need to provide additional arguments why their technology is a great advance compared to previous tools. This could be done in different ways. One possibility that comes to my mind would be pooled CRISPRi screens as suggested by the authors themselves in the discussion section to identify different conditionally essential genes in lambda and P1 (and possibly additional phages). Another possibility would be to show if / that the authors' technology can be applied to phages that are (unlike lambda or P1) highly resilient to DNA-targeting tools like CRISPR-Cas. This is true for many phages including, e.g., well-studied lab phages T4 (due to covalent genome modification) and T5 (unknown mechanism) and could greatly limit the applicability of this technology.

Minor points:

1) The manuscript text in line 126 talks about the mcp gene of P1 in Figure 1 but the corresponding lane in the efficiency-of-plating assay is labeled 23. Even though this is explained in the figure legend I would suggest to directly label the EOP assay mcp instead of 23.

2) The authors show that a knockdown of ral has no phenotype because the indicator strain that they use has no active restriction-modification system (lines 172-174). It is for sure not critical but would be quite cool to show that this gene is conditionally essential for growth on a strain with an active restriction-modification system, highlighting the potential of the authors' technology for the investigation of phage-encoded anti-immunity factors.

Reviewer #3: This manuscript describes a method for performing CRISPRi on phage genes. A plasmid system for the expression of a catalytically inactive Cas12 and guide RNA molecules in E. coli was designed. Genes were targeted in two model E. coli phages, lambda and P1. The authors show that targeting of essential genes led to large reductions in phage replication while targeting non-essential genes generally did not affect replication. The difficulties in using this method on phages are highlighted.

This manuscript clearly demonstrates that the catalytically dead version of Cas12 can block phage replication. This is a result of interest, but not particularly surprising given previously published papers in this area. The problems with this approach for evaluating gene essentiality in phage genomes are important and predictable. Given the large number of polycistronic messages in phage genomes, polarity is bound to be a problem. This manuscript has value in confirming that polarity is a major problem for interpreting this type of experiment performed on phages.

While this manuscript contains data of interest, I do not think that it warrants publication in a journal at the level of PLoS Biology. To warrant publication, the authors would need to provide some mechanistic insight into how this CRISPRi system is functioning on the phage genome. While they hypothesize about the effects of polarity, no experiments are performed to address this issue (e.g. RT-qPCR). As presented, I would conclude that CRISPRi is really not a good method to assess essentiality in phages. This manuscript makes the problems evident, but does not provide insight as to how these problems might be overcome.

---

## [Editor Report · Decision Letter 2]

20 Oct 2023

Dear Dr. Mutalik,

Thank you for your patience while we considered your revised manuscript "Genome-wide CRISPRi knockdown to map gene essentiality landscape in coliphages λ and P1" for publication as a Methods and Resources at PLOS Biology. This revised version of your manuscript has been evaluated by the PLOS Biology editors and the Academic Editor.

Based on our Academic Editor's assessment of your revision, we are likely to accept this manuscript for publication, provided you satisfactorily address the following data and other policy-related requests.

1. DATA POLICY:

A) Supplementary files (e.g., excel). Please ensure that all data files are uploaded as 'Supporting Information' and are invariably referred to (in the manuscript, figure legends, and the Description field when uploading your files) using the following format verbatim: S1 Data, S2 Data, etc. Multiple panels of a single or even several figures can be included as multiple sheets in one excel file that is saved using exactly the following convention: S1_Data.xlsx (using an underscore).

B) Deposition in a publicly available repository. Please also provide the accession code or a reviewer link so that we may view your data before publication.

Regardless of the method selected, please ensure that you provide the individual numerical values that underlie the summary data displayed in the following figure panels as they are essential for readers to assess your analysis and to reproduce it: Figure 5BC.

**Please also ensure that figure legends in your manuscript include information on where the underlying data can be found, and ensure your supplemental data file/s has a legend.**

2. Please provide a blurb which (if accepted) will be included in our weekly and monthly Electronic Table of Contents, sent out to readers of PLOS Biology, and may be used to promote your article in social media. The blurb should be about 30-40 words long and is subject to editorial changes. It should, without exaggeration, entice people to read your manuscript. It should not be redundant with the title and should not contain acronyms or abbreviations.

3. We recommend a change in the title: "Systematic and scalable genome-wide essentiality mapping to identify non-essential genes in phages" or "Systematic and scalable genome-wide essentiality mapping to identify non-essential genes in phages - proof of concept with coliphages λ and P1".

We expect to receive your revised manuscript within two weeks.

*Published Peer Review History*

*Press*

Sincerely,

Paula

---

Senior Editor,

pjaureguionieva@plos.org,

PLOS Biology

---

## [Editor Report · Decision Letter 3]

2 Nov 2023

Dear Dr Mutalik,

Thank you for the submission of your revised Methods and Resources "Systematic and scalable genome-wide essentiality mapping to identify non-essential genes in phages" for publication in PLOS Biology. On behalf of my colleagues and the Academic Editor, Michael Laub, I am pleased to say that we can in principle accept your manuscript for publication, provided you address any remaining formatting and reporting issues. These will be detailed in an email you should receive within 2-3 business days from our colleagues in the journal operations team; no action is required from you until then. Please note that we will not be able to formally accept your manuscript and schedule it for publication until you have completed any requested changes.

PRESS

Sincerely, 

Paula

---

Senior Editor

PLOS Biology
